# Humanoid Head Camera Stabilization Using a Soft Robotic Neck and a Robust Fractional Order Controller

**DOI:** 10.3390/biomimetics9040219

**Published:** 2024-04-07

**Authors:** Jorge Muñoz, Raúl de Santos-Rico, Lisbeth Mena, Concepción A. Monje

**Affiliations:** 1Center for Automation and Robotics, Spanish National Research Council (CSIC), 28049 Madrid, Spain; 2Department of Systems Engineering and Automation, Carlos III University of Madrid, 28903 Madrid, Spain; rauldesantosrico@gmail.com (R.d.S.-R.); lmena@pa.uc3m.es (L.M.); cmonje@ing.uc3m.es (C.A.M.)

**Keywords:** soft robotics, soft robotic neck, camera stabilization, kinematics model, fractional order control

## Abstract

In this paper, a new approach for head camera stabilization of a humanoid robot head is proposed, based on a bio-inspired soft neck. During walking, the sensors located on the humanoid’s head (cameras or inertial measurement units) show disturbances caused by the torso inclination changes inherent to this process. This is currently solved by a software correction of the measurement, or by a mechanical correction by motion cancellation. Instead, we propose a novel mechanical correction, based on strategies observed in different animals, by means of a soft neck, which is used to provide more natural and compliant head movements. Since the neck presents a complex kinematic model and nonlinear behavior due to its soft nature, the approach requires a robust control solution. Two different control approaches are addressed: a classical PID controller and a fractional order controller. For the validation of the control approaches, an extensive set of experiments is performed, including real movements of the humanoid, different head loading conditions or transient disturbances. The results show the superiority of the fractional order control approach, which provides higher robustness and performance.

## 1. Introduction

Camera stabilization during gait is a well known issue in humanoid robotics [1,2,3]. Many artificial vision tasks, as mapping or recognition, depend on image quality aspects like illumination, focus, or image position and rotation. The problem of camera stabilization during the humanoid locomotion lies on the disturbances transmitted from the body motion to the head, which usually holds the cameras and vision sensors. This motion, roughly similar to an inverted pendulum with the pivot point located in the robot’s holding ankle, introduces undesired distortions in the form of image rotations and side shifts.

In the course of humanoid robot gait, various factors contribute to undesirable head motion, leading to potential errors in computer vision applications. Gait disturbances generally arise from the transmission of body motion to the head, primarily manifesting as unwanted rotations and translations along multiple axes. Specific sources may include hip swing, leg lifting, foot placement, and torso tilting during the walking cycle. Consequently, images captured by head-mounted cameras exhibit jitter, affecting tasks dependent on visual data, such as object detection, navigation, or interaction. Therefore, minimizing gait disturbances becomes crucial for enhancing the accuracy of visual perception and overall system reliability.

Understanding the diverse forms of gait disturbances allows researchers to develop effective compensation algorithms tailored to neutralize their effects. At the moment, these issues have been addressed in the literature in two different ways.

The first approach proposes a software image correction. For instance, in [4], a feature tracker and an inertial measurement unit (IMU) are used to estimate the motion between frames, and then, a Kalman filter is used to remove these undesired motions. The proposal in [5] describes the case of a teleoperated M2V2 humanoid robot with binocular video signals fed to the operator remote vision system. In this case, the image is processed to offer a stable vision experience during robot’s gait.

The other approach found in the literature is bioinspiration, which seeks to mimic the solutions found in nature to solve this problem. For instance, birds [6], horses [7], monkeys [8], even small vertebrates [9], and humans [9,10,11] use the same strategy to solve the head stabilization problem. In all these cases, the main corrective action is to perform an opposite motion with the neck in order to cancel the perturbations and maintain the head in a relatively similar inclination. The equilibrium sensors, often located in the head, provide a reference allowing the subject to maintain its head horizontally (and therefore their vision systems) no matter their body positions. The evidences found in these works suggest that head stabilization has an impact in the performance of recognition, tracking and locomotion in the case of these animals, and the bioinspiration approach suggests that it can also be useful in the case of legged robots vision systems. Most authors agree that vertical rotation is typically less critical in terms of the robot’s visual sensory inputs stability, while managing horizontal disturbances plays a vital role in maintaining consistent imagery and facilitating accurate visual processing.

Nevertheless, most of today’s humanoid robots use a two DOF serial mechanism for head orientation, providing a complete scan of their environments, able to carry out frontal (pitch) and vertical (yaw) rotations, but unable to perform lateral bending motion (roll), needed to implement the discussed camera correction (see Figure 1).

The pitch and yaw approach simplicity makes it a very common solution for humanoid head actuators. Robots such as HRP-4 [12], Honda ASIMO [13] or TEO [14] follow this two DOF design (right of Figure 1). Similar motion capabilities are provided by the four-bar mechanism robotic neck in [15], but also restricted to pitch and yaw motions. Because of that limitation, the described mechanical stabilization is not possible in these prototypes; therefore, the software approach is the only option in these cases.

Some designs, like the neck proposed in [16], or the necks of Albert HUBO [17] and Dav [18], have actually three DOF, but even having the motion capabilities, there are no works in the literature regarding head stabilization for these robots. Probably, these designs are conceived to provide more human-like gestures or other tasks, and using their vision systems during locomotion is out of their scope.

Only a few works propose head stabilization implementations. For instance, in [19], an IMU is used to measure the head inclination in order to stabilize the KOBIAN [20] robot’s head, reporting an improvement in the virtual point following task. In [21], a combination of bioinspired software filters and mechanical corrective actions are applied in two combined robot platforms, the biped robot SABIAN (a clone of the robot WABIAN [22]), and the iCub robot’s head [23]. In this case, two head orientation control strategies are also compared, a classic inverse kinematic approach and a bioinspired one based on feedback error learning.

Our humanoid robot TEO (see Figure 2) falls into the first group, featuring a two DOF neck, and therefore unable to perform these corrections either. However, in the last years, the projects HumaSoft (Diseño y Control de Eslabones Blandos para Robots Humanoides, with reference DPI2016-75330-P, funded by the Spanish Ministry of Economics, Industry and Competitiveness, 2016–2021) and SofIA (Articulación blanda inteligente con capacidades de reconfiguración y modularidad para plataformas robóticas, with reference PID2020-113194GB-I00, funded by the Spanish Ministry of Economics, Industry and Competitiveness, 2021–2024) have been working on the design of new soft limbs that can replace the existing rigid link components. Currently, two soft robot prototypes, an arm [24] and a neck [25], have been developed. The bioinspired soft neck was designed to resemble a human neck, able to mimic frontal (pitch) and lateral (roll) movements. Therefore, this soft neck can actually implement the discussed head correction in the robot TEO, as it provides the motions needed to cancel the gait disturbances.

The aim of this work is to propose a solution to the discussed problem of head image stabilization using the soft neck in TEO. For this purpose, the soft neck will replace the current one, and a feedback controller will be used in order to maintain the head orientation at all times, but, given the neck’s nonlinear characteristics (see [26]) and the disturbances expected during operation, it is important to consider a robust control strategy. Two types of controllers are proposed to this end: a proportional integral derivative (PID) controller and a fractional order proportional integral (FOPI) controller. In order to validate this approach, a set of varying mass and disturbances experiments has been designed and implemented. The results clearly show the improvement in performance and stability when using the fractional order controller.

### 1.1. The Humanoid Robot TEO

The robot TEO (Figure 2) is a full size humanoid with 4 limbs of 6 DOF, 2 DOF in its torso, and 2 DOF in the neck, making a total of 28 DOF (hands not included). It has proprioceptive sensors distributed in different locations (IMU, force-torque), and an advanced vision system with high resolution cameras (3D and 2D) located in the head. It was built by the Robotics Lab team http://www.roboticslab.uc3m.es (accessed on 1 March 2024) of Carlos III University of Madrid [14], and designed as a service robot able at the moment to perform challenging tasks such as ironing fabrics [27] or catering [28,29,30].

**Figure 2 biomimetics-09-00219-f002:**
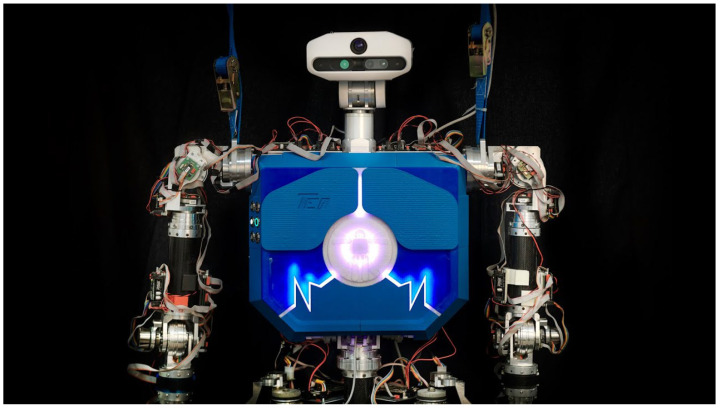
Humanoid robot TEO.

The current TEO’s neck can be observed in Figure 2. Only pitch and yaw motions are possible with this design, as is the case in most humanoid robots found in the literature. That configuration makes the camera image correction strategy described before very difficult to apply. Thus, replacing those 2 DOF by the soft neck described in [25] will grant a more convenient neck configuration space and will provide a solution to that limitation.

### 1.2. Soft Robotic Neck

The soft neck, shown in Figure 3, consists of an thermoplastic elastomer (TPE) central soft link resembling a spine, and a parallel mechanism driven by cables (acting as tendons), which produce a tilt in the upper platform that can be measured by the attached IMU. The three tendons are evenly spaced, showing a separation of 120 deg between each other (see Figure 4).

All parts can be produced with a 3D printer, including the soft link. This neck is designed to handle a payload of 1000 g with an approximate total weight of 20 g (excluding motors and electronics). The three tendons are moved by actuators located at the neck’s base, which in turn are composed by a motor with planetary gear, an encoder and a driver, with the following characteristics:Driver: Technosoft iPOS 3604 MX-CAN. 144 W, 12–36 Volt, 4 Amp.Motor: Maxon RE 16 (118739). Graphite brushes, 48 Volt, 4 Watt.Gear: Maxon GP 16 A (134777). Diameter 16 mm, reduction 24:1.Encoder: Maxon MR (201937). Type M, 512 pulse, 2 Channels.

Each actuator position and velocity low level control is managed by the iPos intelligent motor driver. The actuator absolute position sets the final tendon length, and then, the upper platform rotation. In consequence, their velocities also control the upper platform rotational speed.

According to [31,32], this soft neck is hyper-redundant, and the DOF cannot be considered as usual. Nevertheless, there is a connection between the neck’s final tilt and the actuator positions. Adjusting the appropriate tendon lengths, the end effector can reach any *X* (pitch) and *Y* (roll) target. The *Z* axis rotation (yaw) is not considered, since it cannot change because of the neck design. Two rotations will then define the robot workspace as shown in Figure 3. These final pitch and roll rotations will be considered outputs in order to define a system model.

## 2. Plant and Control System Description

Given the described tendon configuration and the workspace covered by the neck, we are facing a multi-input/multi-output (MIMO) system with a non-linear behavior caused by the robot design and the central soft link, increasing the system modeling complexity. Fortunately, decoupling the variables as proposed in [33] allows modeling this system with two simpler single-input/single-output (SISO) systems as described below.

When a tendon is pulled, the effect produced is a top platform rotation around its particular axis (see Figure 4) and the consequent translation in a perpendicular direction. Note that the tip inclinations and positions are correlated due to the neck mechanics.

Using the simplification described in [33,34], the following input redefinition can be used:(1)$$\theta_{i}\overset{▵}{=}k_{\theta }\left[P_{1}-0.5\left(P_{2}+P_{3}\right)\right],$$
(2)$$\phi_{i}\overset{▵}{=}k_{\phi }\left[0.866\left(P_{2}-P_{3}\right)\right],$$
(3)$$\delta_{i}\overset{▵}{=}\frac{P_{1}+P_{2}+P_{3}}{3},$$
where $\theta_{i},\phi_{i},\delta_{i}$ are the decoupled inputs, that depend on the tendon lengths ($P_{1},P_{2},P_{3}$). Given that link compression can be neglected, as neck geometry favors the lateral bending, δ can be considered constant (in this case, $\delta_{i}=0.12\;m$). The parameters $k_{\theta }$ and $k_{\phi }$ are variable and depend on the current neck position. Since the proposed controllers are robust to plant changes, they can be considered constant for modeling. Furthermore, given the empirical approach, they can be integrated into the identified transfer function, and thus can be defined as $k_{\theta }=1\;\mathrm{rad}/m$ and $k_{\phi }=1\;\mathrm{rad}/m$ without loss of generality.

The decoupling intention is to consider an actuation variable affecting just one of the system outputs. For instance, according to this scheme, the final pitch angle (θ) only depends on $\theta_{i}$, making the other inputs ($\phi_{i},\delta_{i}$) contribution negligible for the pitch angle output.

Replacing the considered $k_{\theta }$ and $k_{\phi }$ values and taking time derivatives in Equations (Equation 1)–(Equation 3), the same decoupling equations can be declared considering actuator velocities as model inputs.
(4)$${\dot{\theta }}_{i}\overset{▵}{=}V_{1}-0.5\left(V_{2}+V_{3}\right),$$
(5)$${\dot{\phi }}_{i}\overset{▵}{=}0.866\left(V_{2}-V_{3}\right),$$
(6)$$0={\dot{\delta }}_{i}\overset{▵}{=}V_{1}+V_{2}+V_{3},$$
where ${\dot{\theta }}_{i},{\dot{\phi }}_{i},{\dot{\delta }}_{i}$ are the decoupled input angular velocities, that depend on the tendon linear speeds ($V_{1},V_{2},V_{3}$).

In this case, the second set of Equations (Equation 4)–(Equation 6) were used; therefore, the considered plant inputs are ${\dot{\theta }}_{i}$ and ${\dot{\phi }}_{i}$ (remember that ${\dot{\delta }}_{i}=0$). In order to define the plant outputs, the tip’s angles (θ, ϕ), which can be directly measured by the IMU, were considered for convenience, introducing a natural integrator in the system.

### System Identification

Using the decoupling scheme defined by Equations (Equation 4) and (Equation 5), the soft neck dynamics can be modeled using two SISO systems. The transfer functions $G_{p}$ and $G_{r}$ will model the measured outputs (θ, ϕ) as a function of the redefined inputs (${\dot{\theta }}_{i}$, ${\dot{\phi }}_{i}$). Therefore, the measured pitch and roll angles are considered as system outputs while the actuator dependent decoupled pitch and roll actions are considered as inputs for identification. These dynamics are subject to a non-linear behavior because of the materials used and the neck design.

Instead of using non-linear tools to model the system and then linearize, it can be modeled with linear tools like the transfer function, as shown in [26,34] if the resulting model accuracy is adequate. During this process, all non-linear behavior existing in the real system is neglected, as the transfer function can only model a linear behavior. Using that simplification allows modeling the system with an identification algorithm like recursive least squares [35].

In order to obtain the model parameters, an identification experiment was performed to capture the plant input-output behavior. For this purpose, a step input was introduced in the plant, and the corresponding output was registered. Given that the system input is a velocity, but the neck inclination has a limited working range, a sign alternating input is needed to avoid the maximum inclination that would cause output saturation. For the same reason, the expected transfer function order is two, including a real pole and an integrator. After testing different modeling options, the most accurate result was obtained with the standard two-pole and gain control engineering model.

The results obtained during the identification process are shown in Figure 5, including the input data (square signal), the measured plant output (real) and the simulated response (model) for the pitch ($G_{p}$) and roll ($G_{r}$) plant models.

A correct model behavior is observed, featuring an integrator, as expected due to the discussed system definition, and evidencing the conversion of velocity inputs to position outputs. The transfer functions obtained through a recursive least squares identification algorithm are: (7)$$G_{p}\left(s\right)=\frac{9.8674}{\left(s+9.906)(s+0.001973\right)},$$
(8)$$G_{r}\left(s\right)=\frac{9.5898}{\left(s+9.688)(s+0.003446\right)}.$$

Modeling accuracies are shown in Table 1 including the Akaike’s final prediction error (FPE) [36] and the mean square error (MSE).

The resulting model shares many similarities with the one presented in [34], as both approaches share several fundamental assumptions about the underlying physics. In this case, as shown below in the experimental section, the simulations match the actual system behavior with high accuracy, validating the proposed model.

Note how two poles are enough to model these systems with excellent accuracy. One of them is a close to 10 rad/s pole, and the other is almost an integrator. The neck’s kinematics reduces the input effect as a function of the current angle, which explains why that pole is not a pure integrator. In essence, the more the tilt, the less the actuator effect on the final position, moving the pole away from the origin. That non-linear behavior can be linearized around any working point across the full operation range.

Given that the system identification was done using data captured over a range of inclinations from 0 deg to 20 deg (see Figure 5), the model obtained is a linearized system with a working point between the neck’s rest point in 0 deg and a 20 deg inclination.

Therefore, slightly different plant dynamics are expected for each neck configuration, which can degrade the specified control performance. In these circumstances, it is important to provide a robust control strategy in order to minimize the impact of plant dynamics changes on the final closed loop system performance. Similar to the literature, we define robustness as the ability to maintain a similar closed loop system response despite changes in plant parameters (usually gain), regardless of disturbance actions or external loads, allowing the system to track a reference with reasonable accuracy.

## 3. Control Strategy

In order to control the pitch and roll neck’s inclination, a feedback loop will be used for reference tracking, and a robust controller to shape the desired neck performance as shown in Figure 6. Controller robustness will be achieved fulfilling specific loop constraints based on a Bode’s ideal transfer function [37], as discussed in [38,39].

The control scheme proposed use integer and fractional order controllers. While the first ones apply derivatives and integrals to determine the control signal based on the measured tracking error, the fractional order controllers extend the capabilities of classical control blocks by generalizing the order of derivative and integral operators, achieving design specifications that their integer counterparts cannot meet [38,40]. Given the fractional controller flexibility, in this case, only one fractional operator is enough to completely fulfill the robust specifications in contrast to the two operators (derivative and integral) required in the integer controller case.

The fractional order proportional integral derivative (FOPID) controller was formulated for the first time by Podlubny [41] and later studied in works such as [42,43,44]. In this case, the specifications can be achieved with a fractional order proportional integral (FOPI) controller, providing an excellent result as shown in simulations and experiments. The FOPI transfer function is:(9)$$FOPI\left(s\right)=k_{p}+k_{a}s^{\alpha }=k\left(1+\tau_{a}s^{\alpha }\right),$$
where $s^{\alpha }$ is the fractional order integral (when α<0) operator.

An approximation of the fractional order operator $s^{\alpha }$ is needed in order to implement the FOPI controller in the feedback control scheme. In this case, the equivalent finite impulse response discrete operator known as Grünwald-Letnikoff approximation described in [45] was used.

A PID controller was also implemented for comparison. The PID transfer function considered is:(10)$$PID\left(s\right)=k_{p}+\frac{k_{i}}{s}+k_{d}s=k\left(1+\frac{1}{\tau_{i}s}+\tau_{d}s\right).$$

Note that both controllers defined in Equations (9) and (10) have the same number (three) of adjustable parameters. A robust behavior can also be achieved with an integer controller [46,47], but this is only applicable for very specific plant and controller conditions, as discussed in [39]. Even when the same number of adjustable parameters are available, integer order controllers are limited to certain specifications. In this case, the PID was unable to fulfill the robustness conditions, and therefore the constraints were relaxed in order to get a feasible system.

### 3.1. Control Specifications

In order to achieve the desired performance, phase margin ($\phi_{m}$) and gain crossover frequency ($\omega_{gc}$) design specifications are imposed as detailed in [48]. According to that method, these open loop variables are directly related to the closed loop performance specifications of damping ratio (ξ) and peak time ($t_{p}$) through the following equations:(11)$$\tan\left(\phi_{m}\right)=\frac{2\xi }{\sqrt{-2\xi^{2}+\sqrt{1+4\xi^{4}}}},$$
(12)$$\omega_{bw}\;\cdot \;t_{p}\;\cdot \;\sqrt{1-\xi^{2}}=\pi \sqrt{1-2\xi^{2}+\sqrt{4\xi^{4}-4\xi^{2}+2}}.$$

Choosing $\phi_{m}=50\;\deg$ and solving Equation (11), a damping ratio of ξ=0.4777 is found, resulting in an expected overshoot percentage of 18.1%. Then, trying to get a peak time of one second, similar to the human neck response (see [49]) and fast enough to solve the stabilization problem proposed, a closed loop bandwidth of $\omega_{bw}=4.65\;\mathrm{rad}/s$ is found using Equation (12). Then, according to [48], the required open loop crossover frequency should be 2.87 rad/s (rounded up to 3 rad/s for convenience). Therefore, the open loop frequency specifications are $\phi_{m}=50\;\deg$ and $\omega_{gc}=3\;\mathrm{rad}/s$, and the expected behavior is a time response of $t_{p}=0.96\;s$ with 18.1% overshoot.

### 3.2. Controller Design

Classic tuning methods cannot be applied to fractional order controllers, but other techniques have been developed. For instance, the works in [44] propose the numeric solution of the non-linear equation systems in order to find the controller parameters fulfilling the specifications. In a similar approach, other works propose the use of optimization methods such as Particle Swarm Optimization [42,50], or Artificial Bee Colony algorithm [51] in order to find these parameters. A comparative study for optimization algorithms applied to fractional controller tuning can be found in [52].

The graphical solution of the non-linear equations is another approach developed recently. For instance, the works in [53] or [54] show a very intuitive and straightforward solution to the tuning problem. Given the proposed FOPI controller, the iso-m method described in [54] can provide the controller parameters by following a series of simple steps that do not require heavy computational efforts. Only the model’s frequency response shown in Figure 7 is required to apply that tuning method.

The following operations were carried out for the iso-m controller tuning in the pitch case:The plant model phase ($\Phi_{p}=-106.7\;\deg$) and phase slope ($m_{p}=-36.5\;\deg/\mathrm{decade}$) were found through the model transfer function (Equation 7) at the crossover frequency ($\omega_{gc}=3\;\mathrm{rad}/s$) (see Figure 7).In order to achieve the phase margin specification, the controller need a phase of $\Phi_{c}=\left(-\left(-106.7\right)+50-180\right)\;\deg$ at $\omega_{gc}$, that is, $\Phi_{c}=-23.3\;\deg$. Besides, the controller is required to contribute with the opposite phase slope, that is, $m_{c}=36.5\;\deg/\mathrm{decade}$.Based on the values computed in Step 2, the fractional order (exponent) obtained in the slopes graph, available in [54], is α=−0.85.Having α=−0.85 and $\Phi_{c}=-23.3\;\deg$, $\tau_{a}$ is obtained as:$\tau_{x}=\left[\sin(\alpha \pi /2)/\tan(\Phi_{C})\right]-\cos\left(\alpha \pi /2\right)$,$\tau_{a}=1/\left[\tau_{x}{\omega_{gc}}^{\alpha }\right]$,resulting $\tau_{a}=1.2542$.Finally, the controller gain *k* is computed in order to adjust the crossover frequency,$k=1/|\bar{C}\left(j\omega_{cg}\right)G\left(j\omega_{cg}\right)|$,resulting k=2.5773.According to Equation (9), the controller parameters are: $k_{p}=2.5773$, $k_{a}=3.2325$, α=−0.85.

Following the same procedure for the roll plant model, the controller parameters $k_{p}=2.6299$, $k_{a}=3.2395$, α=−0.86 were found. A summary with all the controller parameters is shown in Table 2.

The controller transfer functions are then: (13)$$\begin{array}{c} FOPI_{p}\left(s\right)=2.5773+3.2325\;\cdot \;s^{-0.85}, \end{array}$$(14)$$\begin{array}{c} FOPI_{r}\left(s\right)=2.6299+3.2395\;\cdot \;s^{-0.86}. \end{array}$$

The Bode diagram shown in Figure 8 displays how the resulting controllers meet the specifications proposed. Coincident with the 3 rad abscissa, values of 0 dB for the magnitude and a phase margin of 50 deg are observed, which implies that both loops fulfill the desired specifications. A zero phase slope can also be found at the crossover frequency, as expected from the iso-m tuning method description. This flat phase around the crossover frequency guarantees a constant phase margin (and therefore a constant overshoot behavior) around its nominal value, providing system robustness [38].

Due to the fractional nature of the integrator considered, whose phase contribution is −90 · α=−76.5 deg rather than the −90 deg of an integer order operator, the phase margin at low frequencies is nearly 20 deg. Therefore, the expected result is a reasonably stable system for any value of gain, no matter how small.

Once the controllers are designed, time response simulations for different gain values were performed. The results are shown in Figure 9.

If the system’s gain do not change, the peak time is perfectly in line with expectations (0.989 s). In the case of different gains, the response is faster (higher gain) or slower (lower gain), but the system stability is the same, as shown through by the constant peak values (iso-damping property). Only a slightly higher overshoot of 27% is obtained compared to the initial specifications (18.1%), but that is probably caused by the operator approximation used in the simulation, as will be shown in the experiments which perfectly fit the specifications.

Once the fractional order controllers are solved, the integer order PID tuning was addressed. The PID controller design has been widely discussed in the control literature [55], resulting in a wide variety of different methodologies for controller parameter tuning. In this case, an adapted version of the algorithm described in [54] has been used to obtain the parameters $k_{p}$, $k_{i}$ and $k_{d}$ fulfilling the same specifications of phase margin and the gain crossover frequency. This modified version, described in [39], allows defining a slope in the range of possible values that the integer order controller can reach. In this case, a flat phase slope was impossible, leaving two possible options: 1. Relax the flat slope constraint and loose the robustness properties, or 2. Relax the specifications and keep the robustness. We decided to take the second option for the sake of a better comparison, and increased the phase margin to $\Phi_{m}=70\;\deg$. This allows compliance with the requirement described in [39] of $m_{c}>\left|sin\left(2\Phi_{c}\right)\;\cdot \;log\left(10\right)/2\right|$, while improving the PID stability conditions.

The following operations must be carried out for the counter-slope PID controller tuning method in the pitch case:The plant model phase ($\Phi_{p}=-106.7\;\deg$) and phase slope ($m_{p}=-36.5\;\deg/\mathrm{decade}$) were found through the model transfer function (Equation 7) at the crossover frequency ($\omega_{gc}=3\;\mathrm{rad}/s$) (see Figure 7).In order to achieve the phase margin specification, the controller needs a phase of $\Phi_{c}=\left(-\left(-106.7\right)+70-180\right)\;\deg$ at $\omega_{gc}$, that is, $\Phi_{c}=-3.3\;\deg$. Besides, the controller is required to contribute with the opposite phase slope, that is, $m_{c}=36.5\;\deg/\mathrm{decade}=0.638\;\mathrm{rad}/\mathrm{decade}$.Based on the values computed in Step 2, the variables *T* and *M* must be computed using the following equations:$T=tan(\Phi_{c})$$M=m_{c}\;\cdot \;\left(1+T^{2}\right)/log\left(10\right)$The results obtained in the pitch controller case are T=−0.057 and M=0.278.Using the previous values, the controller parameters can be obtained through the equations:$\tau_{d}=\left(M+T\right)/\left(2\omega_{gc}\right)$$\tau_{i}=\left(2/\omega_{gc}\right)/\left(M-T\right)$The resulting parameters obtained for the pitch PID controller are $\tau_{d}=0.037$ and $\tau_{i}=1.99$.Finally, the controller gain *k* is computed in order to adjust the crossover frequency, resulting k=3.126.According to Equation (10), the controller parameters are: $k_{p}=3.126$, $k_{i}=1.57$, $k_{d}=0.1151$.

Following the same procedure for the roll plant model, similar controller parameters $k_{p}=3.152$, $k_{i}=1.583$, $k_{d}=0.122$ were found. A summary with all the controller parameters is shown in Table 3.

The PID transfer functions are then: (15)$$\begin{array}{c} PID_{p}\left(s\right)=3.126+\frac{1.57}{s}+0.115\;\cdot \;s, \end{array}$$(16)$$\begin{array}{c} PID_{r}\left(s\right)=3.152+\frac{1.583}{s}+0.122\;\cdot \;s. \end{array}$$

The Bode diagram shown in Figure 10 displays how the resulting controllers meet the specifications. Coincident with the 3 rad abscissa, values of 0 dB for the magnitude and a phase margin of 70 deg are observed, which implies that both loops fulfill the desired specifications. A zero phase slope can also be seen at the crossover frequency, as expected.

Compared to the previous controller, a smaller phase margin can be observed for the low frequencies around $3\times 10^{-2}\;\mathrm{rad}/s$, meaning that the plant parameter changes will have an increased impact in the system’s stability.

Once the controllers are designed, time response simulations for different gain changes were performed. The results are shown in Figure 11.

If the system’s gain do not change, the peak time and predicted overshoot is perfectly in line with expectations. In the case of different gains, the response time change as expected, but there is a poor robust behavior, as evidenced by the changing overshoot values. Note that the specifications were relaxed in this case, increasing the phase margin, which should lead to an improved stability, but due to the reduced phase margin at low frequencies, the behavior tends to be unstable for low gains. In the following section, the previously discussed issues will be experimentally tested and validated.

## 4. Results and Discussion

The first experiment performed is a step input of 0.17 rad(9.74 deg) introduced as a reference in the closed loop system. All the angular positions were recorded during the experiment for different masses attached to the neck’s tip. The results obtained for the pitch and roll control are shown in Figure 12 and Figure 13, respectively, for both PID and FOPI controllers.

The first impression is that both systems are quite robust to mass changes. The curves show that overshoots are really similar as expected from the specifications. Stability is excellent despite the different masses applied in the fractional controller case, and a slightly faster response ($t_{pf}\approx 1.7\;s$ vs. $t_{pi}\approx 2.0\;s$) compared to the PID controller can be observed. In the PID case, a worse robustness behavior can be observed. Note the higher overshoot variation and the glitch between t=0 s and t=0.5 s (see Figure 13). Although the robustness is quite good in both controllers, making the curves overlap, a detailed study of the results show that the PID unstable behavior increase with the mass, unlike the FOPI controller which keeps the same stability parameters throughout the entire experiment.

Once verified the correct neck behavior, the next experiment aims to evaluate the proposed head stabilization system performance. For this purpose, the soft neck was installed on the humanoid’s trunk using the former neck mount, and then several motions were performed in order to recreate the disturbances associated with the walk. A total weight of 1000 g was attached to the neck in order to consider the worst case scenario.

The pitch and roll targets were set constant during all the experiment time, both to zero inclination, in order to keep the camera horizontally stabilized. At the same time, the frontal and axial humanoid trunk pose was changed at 8-s intervals for five different positions (see Table 4). Given the neck setup, these angles are fully transmitted to the neck base, forcing a tilt correction. The experiment was performed using both control strategies in order to compare their stabilization and performance characteristics. The IMU measurements were captured during the experiment, showing any deviation from the horizontal position (0, 0) and the transient system response. These head pitch and roll measured values are shown in Figure 14 for the whole experiment.

As observed in the figure, inclination changes correspond to the time sequence proposed, appearing in 8-s intervals. In all cases, an opposite angle is obtained at the beginning, due to the fast trunk position change, followed by a ramp, which results from the combination of the corrective action and the trunk movement. Finally, the inclination values tend to zero because of the neck final compensation.

Although both systems give roughly similar results, a closer look at the figure shows that PID stability is much worse, leading to sustained oscillations, indicative of an almost exhausted phase margin, which is to be expected in view of the Bode diagrams shown in Figure 7. The mass increase produces a lower loop gain that shifts the crossover frequency towards lower abscissa values, where a much smaller phase margin is available, making the system less stable. The FOPI controller performance is much cleaner, without notable oscillations. Note that the largest error (3.5 deg) only lasts several seconds, and its value is three times lower than the tilt caused by the trunk (12 deg).

Using the same setup, a stability test against external disturbances was carried out. A research technician proceeded to push and move the neck with his hand (Figure 15), forcing it to move out of the rest position (pitch and roll at zero) while the control system was in operation and the neck loaded with a weight of 1000 g. In this way, we can study the controller’s behavior in the face of a strong shock or disturbance.

The experiment result is shown in Figure 16. As observed in the response, stability recovery with the FOPI controller is much faster (around 5 s), while in the PID controller case the oscillation is permanent due to the small phase margin, as discussed.

Finally, the last experiment considered is a demonstration of the system’s performance while stabilizing the image of a real camera. Having determined which of the two controllers is more convenient to implement the system, we prepared the FOPI control scheme and installed a camera at the neck’s end effector in order to visualize the stabilization results. In this case, an Intel RealSense was used to record how it would look from the camera’s point of view, while the trunk moves in the same positions discussed above. The camera setup is shown in Figure 17.

Thanks to the FOPI controller, the movements carried out by the neck based on the changes made by the trunk are smooth, allowing a correct stabilization of the camera that maintains its pitch and roll values really close to zero.

All the experiments and results videos are available in: https://vimeo.com/756294204 (accessed on 1 March 2024).

## 5. Conclusions

This work proposes a bioinspired camera stabilization strategy for the humanoid robot TEO. As a difference from the existing literature, our approach is based on a soft neck, since its motion capabilities are more convenient for this task, and provide a better compliance, safer human-robot interaction and adaptability to complex environments.

In this case, the head stabilization is achieved with a feedback control scheme based on the neck’s embedded IMU device, providing full disturbance rejection and effective inclination control, very similar to behaviors observed in the nature. The system allows setting any head gaze and maintain it despite the robot’s lower body positions or gait disturbances, in a way very similar to humans and other animals, which interestingly are also walkers.

Along with the advantages provided by the soft robotics approach, there are also many open challenges, like the problematic kinematic definition or the high non-linearity, which complicates modeling and makes these systems difficult to control. Following the decoupled approach described in [33,34], and a feedback control strategy, an efficient solution using PID and FOPID controllers is proposed.

In order to assess the system performance with both controllers, an extensive set of experiments was carried out, including step response and real humanoid motion tests, all of them under varying payload conditions. As expected, given the plant characteristics, the classic PID controllers provide a poor performance even using robust design constraints, resulting in oscillations and unstable behavior for different inclinations and payloads. However, with the robust fractional controller, the performance and stability are drastically improved. As the theory and experiments avail, the FOPI provides more flexibility and allows achieving more demanding control requirements, offering better results due to a more moderate phase contribution than the integer order operator-based controllers.

## Figures and Tables

**Figure 1 biomimetics-09-00219-f001:**
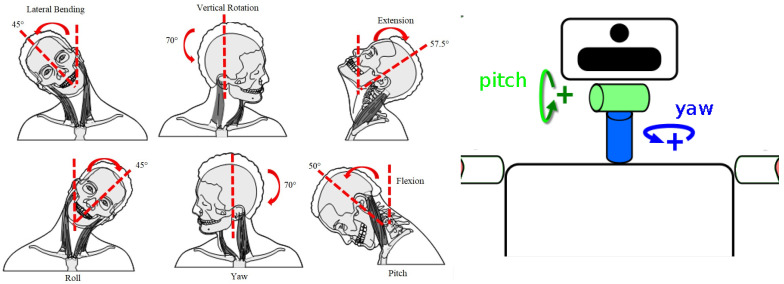
Comparison of the usual neck movements of a human (**left**) and a humanoid robot (**right**).

**Figure 3 biomimetics-09-00219-f003:**
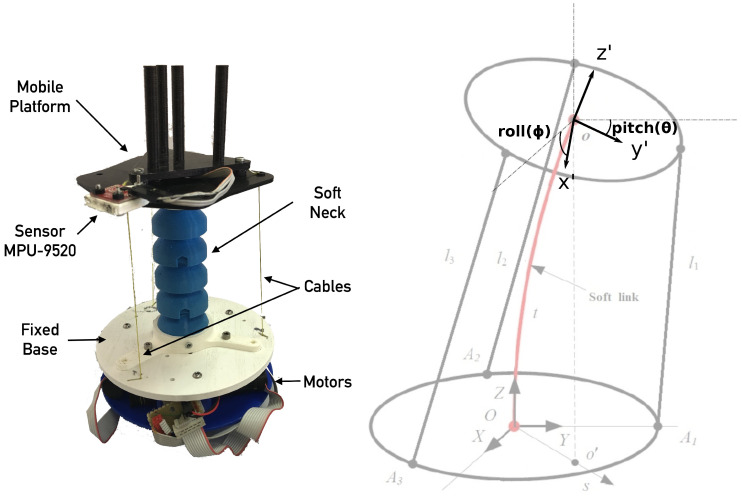
Soft neck platform.

**Figure 4 biomimetics-09-00219-f004:**
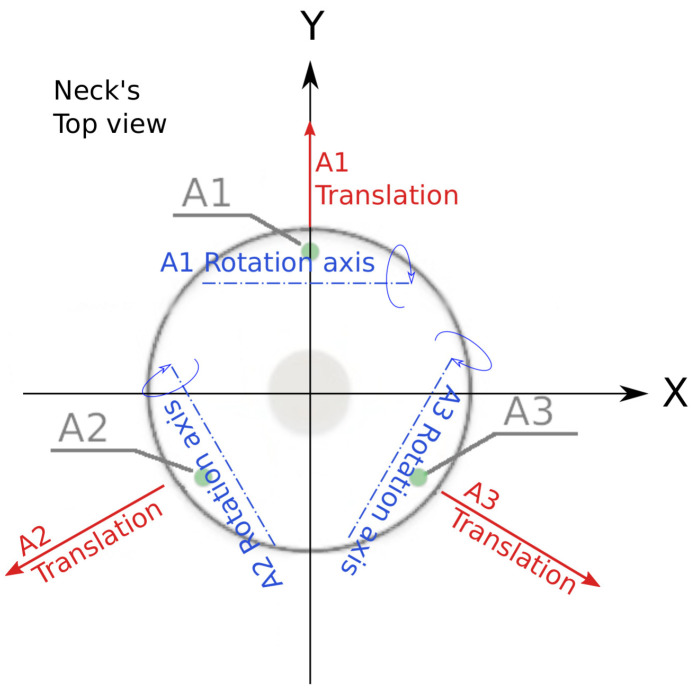
Soft neck tendon actuation effects.

**Figure 5 biomimetics-09-00219-f005:**
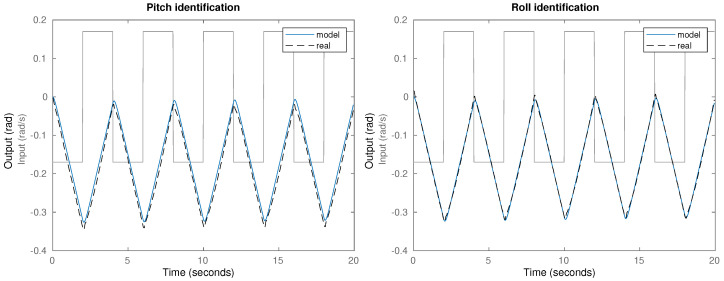
System identification experiment and results for the pitch (**left**) and roll (**right**) transfer functions.

**Figure 6 biomimetics-09-00219-f006:**
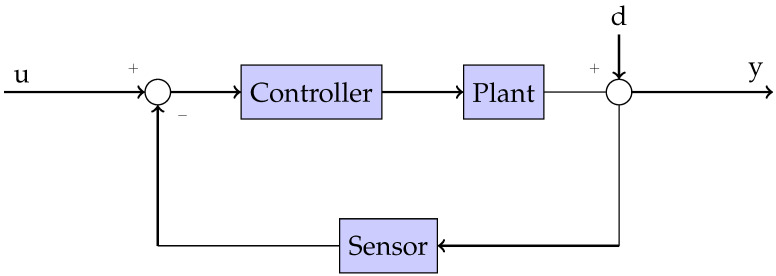
Proposed feedback control loop.

**Figure 7 biomimetics-09-00219-f007:**
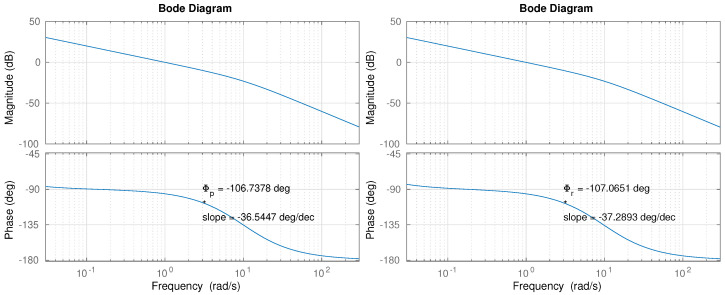
Bode diagram for pitch (**left**) and roll (**right**) plant models.

**Figure 8 biomimetics-09-00219-f008:**
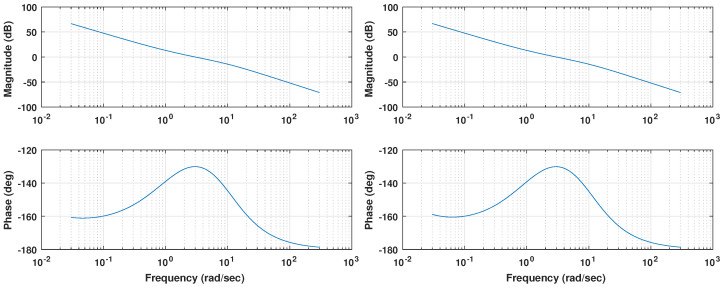
Bode diagram for pitch (**top**) and roll (**bottom**) control loops for the FOPI case.

**Figure 9 biomimetics-09-00219-f009:**
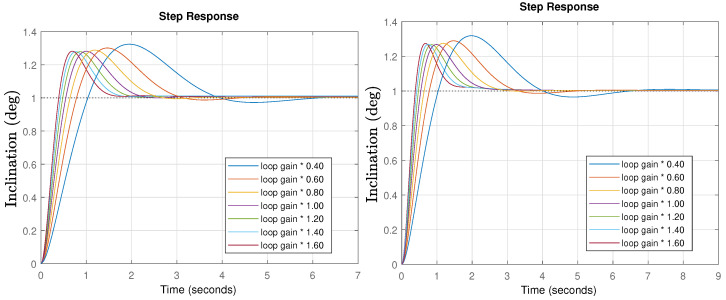
Time response simulation for the pitch (**left**) and roll (**right**) final controlled systems for different loop gains for the FOPI case.

**Figure 10 biomimetics-09-00219-f010:**
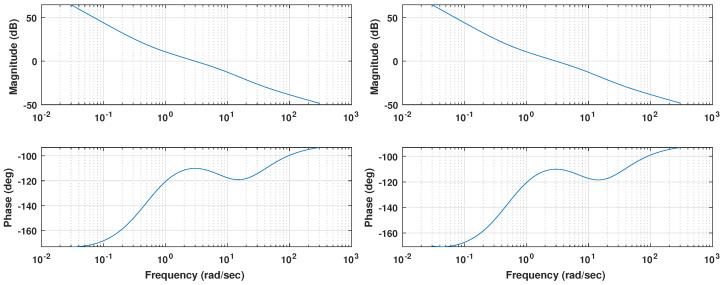
Bode diagram for pitch (**top**) and roll (**bottom**) control loops for the PID case.

**Figure 11 biomimetics-09-00219-f011:**
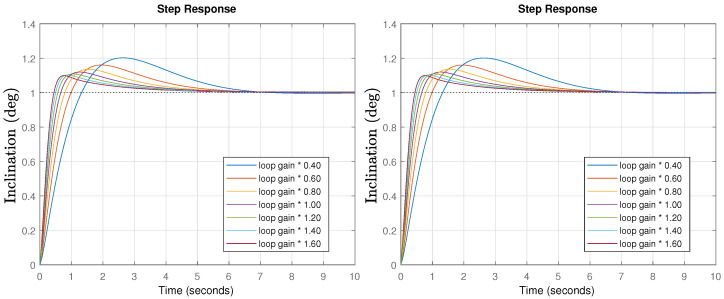
Time response simulation for the pitch (**left**) and roll (**right**) final controlled systems for different loop gain situations for PID case.

**Figure 12 biomimetics-09-00219-f012:**
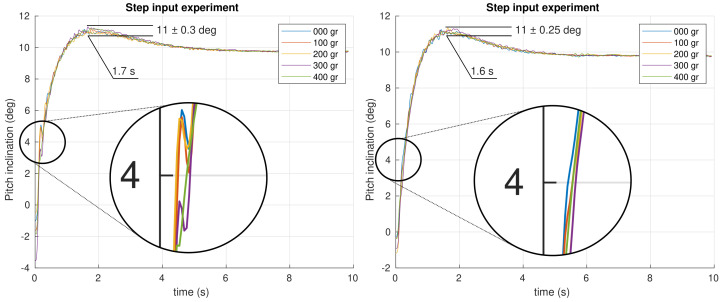
Time response experiment for the pitch control system using PID (**left**) and FOPI (**right**) control schemes.

**Figure 13 biomimetics-09-00219-f013:**
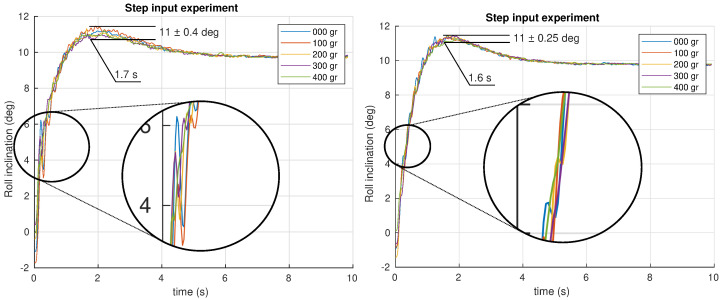
Time response experiment for the roll control system using PID (**left**) and FOPI (**right**) control schemes.

**Figure 14 biomimetics-09-00219-f014:**
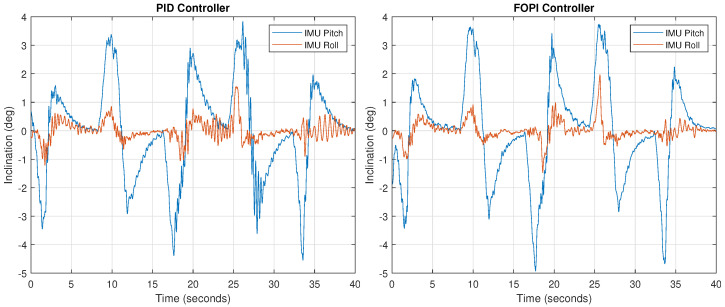
Final stabilized neck pitch and roll angles during the trunk movements of the robot using a PID controller (**left**) and a FOPI controller (**right**).

**Figure 15 biomimetics-09-00219-f015:**
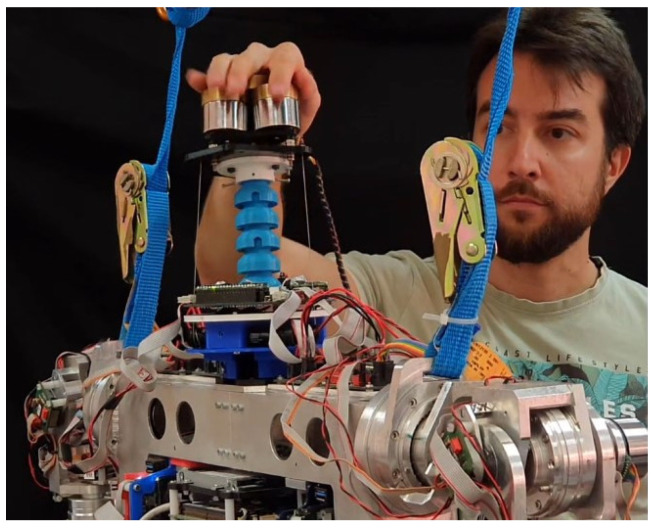
A research technician pushes the neck to study the behavior of both controllers and check their robustness to significant disturbances.

**Figure 16 biomimetics-09-00219-f016:**
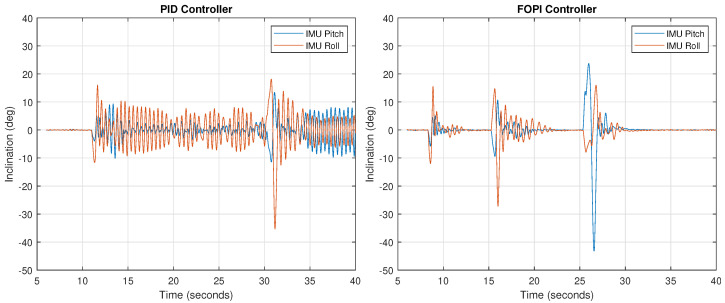
Stabilization of the neck using a PID controller (**left**) and a FOPI controller (**right**) during disturbances generated by a person pushing it.

**Figure 17 biomimetics-09-00219-f017:**
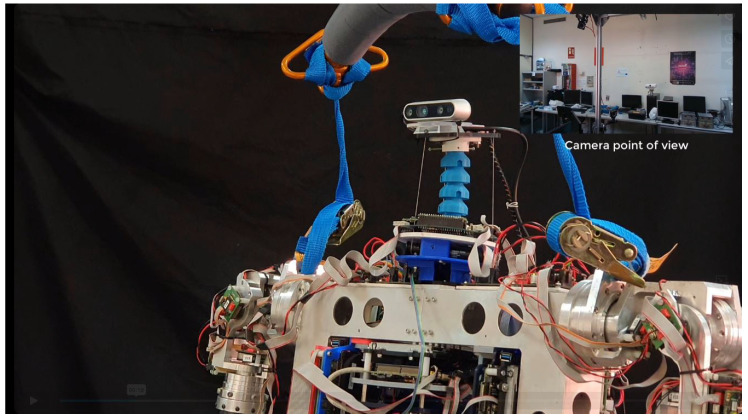
Stabilization of a RealSense camera on a soft neck installed on the humanoid robot TEO while performing trunk movements.

**Table 1 biomimetics-09-00219-t001:** Model accuracy for pitch and roll transfer functions.

Model	Fit to Data	FPE	MSE
$G_{p}$	96.23%	1.339 × 10^−5^	1.326 × 10^−5^
$G_{v}$	95.67%	1.747 × 10^−5^	1.730 × 10^−5^

**Table 2 biomimetics-09-00219-t002:** FOPI controller parameters for $\phi_{m}=50\;\deg$ and $\omega_{gc}=3\;\mathrm{rad}/s$ for pitch and roll FOPI control strategies.

Controller	$k_{p}$	$k_{a}$	α
Pitch	2.5773	3.2325	−0.85
Roll	2.6299	3.2395	−0.86

**Table 3 biomimetics-09-00219-t003:** PID controller parameters for $\phi_{m}=70\;\deg$ and $\omega_{gc}=3\;\mathrm{rad}/s$ for pitch and roll PID control strategies.

PID Controller	$k_{p}$	$k_{i}$	$k_{d}$
Pitch	3.126	1.57	0.115
Roll	3.152	1.583	0.122

**Table 4 biomimetics-09-00219-t004:** Trunk angles during the stabilization experiment in degrees.

Pose	p1	p2	p3	p4	p5	Units
Frontal	0	−12	12	−12	12	deg
Axial	0	20	20	−20	−20	deg
time	0	8	16	24	32	s

## Data Availability

All data and codes associated with this work is available here: https://doi.org/10.21950/TAIGC4.

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
