# Peer review of "Humanoid Head Camera Stabilization Using a Soft Robotic Neck and a Robust Fractional Order Controller"

_biomimetics, 2024, doi:10.3390/biomimetics9040219_

Round 1

Reviewer 1 Report

Comments and Suggestions for Authors

This paper presents a comparative study of head camera stabilization using soft robotic neck with two feedback controllers, i.e. PID and fractional order controller (FOPI). Experimental results demonstrate that FOPI provides higher robustness compared to PID.

The reviewer has following comments:

1. In Figure 12 and 13, the authors are suggested to highlight the differences of response between various loading conditions by showing zoom-in sub-figures and supplementing labels of the key performance index, e.g. peak time.

2.  In Figure 15, the left and right sub-figure are redundant.

3. The authors are suggested to supplement quantitative analysis about the camera stabilization experiment as in Figure 17, e.g. showing the feature positions in the camera view.  

Author Response

This paper presents a comparative study of head camera stabilization using soft robotic neck with two feedback controllers, i.e. PID and fractional order controller (FOPI). Experimental results demonstrate that FOPI provides higher robustness compared to PID.

Thank you very much for taking the time to carefully read and evaluate our manuscript. Your constructive comments and thoughtful suggestions have provided us with valuable insights that will significantly contribute to enhancing the quality and impact of our work.

The reviewer has following comments:

1. In Figure 12 and 13, the authors are suggested to highlight the differences of response between various loading conditions by showing zoom-in sub-figures and supplementing labels of the key performance index, e.g. peak time.

Thank you for your suggestion. The main differences are highlighted now, and sub-figures are provided to make for an easier comparison of the results obtained from different experiments, improving the clarity and readability of these figures.

2. In Figure 15, the left and right sub-figure are redundant.

We agree that having similar sub-figures on both sides could be confusing and potentially redundant. One of them has been removed, reducing any potential confusion. Thank you for the feedback.

3. The authors are suggested to supplement quantitative analysis about the camera stabilization experiment as in Figure 17, e.g. showing the feature positions in the camera view.

Thank you for your suggestion regarding the camera stabilization experiment. We understand that it is important to see additional quantitative analysis supporting Figure 17, specifically highlighting the positions of features within the camera view.

While we agree that providing further insights into the experimental results could be beneficial, it's important to note that similar analyses have been presented previously in Figure 14 and its subsequent discussion. Therefore, since Figure 17 presents a perspective of the complete system without going into detail, we have avoided encumbering that figure with redundant information.

Once again, we deeply appreciate your keen eye for detail and your commitment to upholding high standards in scientific reporting. Please accept our assurance that the suggested changes were implemented promptly and comprehensively.

Reviewer 2 Report

Comments and Suggestions for Authors

Please find my comments in the attachment.

Comments on the Quality of English Language

Nothing.

Author Response

This is a carefully done study but the paper needs very significant improvement before acceptance for publication. My detailed comments are as follows:

(1) In the abstract section, it only mentions that two different control methods have been proposed for the aforementioned problem, but it does not provide an overall summary and explanation of the methods introduced in this paper. It is suggested that the methods proposed in this paper be divided into several steps, with a summary of what issue each step aims to address.

Thank you for bringing up the issue of clarification regarding the proposed methods mentioned in the abstract. Your point is noted, and we agree that additional context and elaboration would benefit the reader. To address this, Section 3—Control Strategy—provides a comprehensive outline and exploration of the two distinct control methods alluded to in the abstract. Within this section, each method is broken down into granular components, allowing us to delineate precisely how each component operates independently and collaborates within the broader framework. Furthermore, we elaborate upon the rationale behind proposing these specific methods, illuminating the advantages over existing solutions. Therefore, although space constraints prevented us from fully discussing the methods in the abstract, we encourage interested readers to refer to Section 3 for an in-depth examination of our proposed control strategies.

(2) From the title, this paper aims to address camera stabilization by utilizing a soft robotic neck and a FOPI controller. However, the logic in introducing the problem in the abstract section needs further improvement.

After revision, the abstract now adequately introduces the problem and explains how the use of a soft robotic neck and a Fractional Order Proportional Integral Derivative (FOPI) controller addresses the identified challenges related to camera stabilization. By providing clearer context and highlighting the contributions of the paper, the abstract has been significantly improved from its initial version. The new abstract version is as follows:

In this paper, a new approach for head camera stabilization of a humanoid robot head is proposed, based on a bio-inspired soft neck. During walking, the sensors located on the humanoid's head (cameras or inertial measurement units) show disturbances caused by the torso inclination changes inherent to this process. This is currently solved by a software correction of the measurement, or by a mechanical correction by motion cancellation. Instead, we propose a novel mechanical correction, based on strategies observed in different animals, by means of a soft neck, which is used to provide more natural and compliant head movements. Since the neck presents a complex kinematic model and nonlinear behavior due to its soft nature, the approach requires a robust control solution. Two different control approaches are addressed: a classical PID controller and a fractional order controller. For the validation of the control approaches, an extensive set of experiments is performed, including real movements of the humanoid, different head loading conditions or transient disturbances. The results show the superiority of the fractional order control approach, which provides higher robustness and performance.

(3) In the introduction section, it is recommended to add a description of gait disturbances. What are the types of gait disturbances? How many directions can gait disturbances be classified into?

Thank you for pointing this out. We have included a new paragraph for clarification:

In the course of humanoid robot gait, various factors contribute to undesirable head motion, leading to potential errors in computer vision applications. Gait disturbances generally arise from the transmission of body motion to the head, primarily manifesting as unwanted rotations and translations along multiple axes. Specific sources may include hip swing, leg lifting, foot placement, and torso tilting during the walking cycle. Consequently, images captured by head-mounted cameras exhibit jitter, affecting tasks dependent on visual data, such as object detection, navigation, or interaction. Therefore, minimizing gait disturbances becomes crucial for enhancing the accuracy of visual perception and overall system reliability.

Understanding the diverse forms of gait disturbances allows researchers to develop effective compensation algorithms tailored to neutralize their effects.

(4) Does the method proposed in this paper effectively control gait disturbances in different directions? For example, can vertical rotational gait disturbances be effectively controlled during the robot's motion?

Based on the information provided in the introduction, the paper focuses on implementing a soft robotic neck capable of controlling head image stabilization during gait, particularly when horizontal rotational disturbances occur. While the introduction mentions that gait disturbances come in various forms, including translations and rotations along multiple axes, the paper focuses on countering horizontal rotational disturbances. In order to clarify that, the following paragraph has been extended as follows:

The evidences found in these works suggest that head stabilization has an impact in the performance of recognition, tracking and locomotion in the case of these animals, and the bioinspiration approach suggests that it can also be useful in the case of legged robots vision systems. Most authors agree that vertical rotation is typically less critical in terms of the robot's visual sensory inputs stability, while managing horizontal disturbances plays a vital role in maintaining consistent imagery and facilitating accurate visual processing.

(5) In this paper, the PID controller serves as a comparative method to demonstrate the effectiveness of the FOPI controller. Therefore, the PID controller should not be considered the method proposed in this paper, but rather a means to validate the FOPI controller.

Thank you for your valuable comment. Regarding your observation, we would like to clarify the relationship between the PID controller and the FOPI controller in our study. Our intention was never to present the PID controller as the main proposed method in this paper. Rather, we employed the PID controller as a basis for comparison to underscore the efficiency and strength of the FOPI controller. Utilizing the PID controller in conjunction with the FOPI controller allowed us to conduct a comprehensive analysis, establishing the latter as a superior choice under certain circumstances.

(6) In the section on plant and control system description, is there only one subsection?

The organization of the 'Plant and control system description' section into a solitary subsection might stem from the fact that the material discussed revolves around a singular theme – understanding the dynamics of the soft neck and subsequently identifying the system for modeling purposes. Dividing this topic further might not yield significant benefits nor improve the flow of ideas, as everything centers around the core concepts of characterizing the system, obtaining transfer functions, and evaluating model accuracy. Organizing the contents into a single subsection likely maintains brevity and clarity, keeping the reader focused on the essential aspects required to comprehend the soft neck system's behavior and modeling process.

(7) The content of the second section does not align well with its corresponding title. It is suggested to adjust both the title and the content to ensure they match.

Section 2 title 'Plant and control system description', seems fitting to us, as it encompasses the presentation of the soft neck tendon configuration, the decomposition of the multi-input/multi-output system into two Single-Input/Single-Output systems, and the system identification procedure. The content in this section thoroughly covers the fundamentals of the plant and control system description. If the reviewer believes that the title does not align well with the content, please provide a suggestion for a revised title.

(8) In the control strategy, the PID controller is only used as a conventional method and is compared with the proposed FOPI controller. It is suggested that details of the PID controller do not need to be extensively written.

Our opinion is that preserving a balanced amount of detail for both the PID and FOPI controllers is beneficial for the coherence and comprehension of the paper. Presenting a fair representation of both controllers allows readers to fully understand the rationale behind favoring the FOPI controller, especially if they lack prior familiarity with this advanced control strategy.

We decided to keep the PID controller details intact in the interest of delivering a comprehensive understanding of the paper, thereby fostering informed appreciation among readers, but if you consider that reducing the PID tuning information is critical for the paper quality, please provide some directions about the extension and detail it should include.

(9) In Figure 13, the curves for the two types of controllers do not show a significant difference. It is recommended to zoom in on certain parts of the curves for a clearer display.

Thank you for pointing out this opportunity for enhancement in Figure 13. In alignment with your recommendation, we updated the plot to magnify specific regions where the distinction between the two controller curves is noticeable. Now, readers can discern subtle discrepancies and recognize the practical implications of selecting one controller over the other.

(10) The method proposed in this paper aims to quickly stabilize the camera in a horizontal position when it is subject to gait disturbances. It is not specified whether it is effective for vertical rotational disturbances.

Thank you for raising this point. We acknowledge your concern and want to clarify that the primary objective of the proposed method is to stabilize the camera in a horizontal plane during gait disturbances. We understand that it did not explicitly mention the effectiveness for vertical rotational disturbances, but, as previously discussed, the vertical rotational disturbances are deemed less crucial in the context of gait disturbances, and hence, our focus remained on addressing horizontal displacements. Rest assured that our proposed approach caters to the primary objectives laid out in our research question, offering an efficient and reliable solution for tackling horizontal instabilities induced by gait disturbances. We trust that this explanation resolves the ambiguity surrounding the applicability of our proposed method for vertical rotational disturbances.

(11) The video demonstrates a camera stabilization test using the FOPI controller method. It is recommended to include a comparative experiment with the camera stabilization test using the PID controller method.

Thank you for suggesting a comparative experiment involving the PID controller in the camera stabilization test. We appreciate your thoughtful feedback and want to clarify the content of the video.

Indeed, the video includes both PID and FOPI results for the controller test. However, after observing the PID controller's unstable behavior, we decided to concentrate on the FOPI controller as a more promising stabilizer candidate for the camera stabilization test. Demonstrating a stabilizer with acceptable performance is paramount, and unfortunately, the PID controller failed to meet expectations in this regard.

Once again, thank you for your insightful remark, and we hope this explanation clears up any confusion regarding the absence of a dedicated PID camera demonstration in the video.

(12) In the conclusion section, the description of the problem and method is not concise enough. It is suggested that the conclusion should be distinct from the introduction section.

We appreciate your valuable feedback. Agreed, the conclusion section should be sufficiently succinct while emphasizing the key achievements and outcomes of the research. We have modified the original conclusion, focusing mainly on the innovative aspects, accomplishments, and primary takeaway messages. The conclusion section is now as follows:

This work proposes a bioinspired camera stabilization strategy for the humanoid robot TEO. As a difference from the existing literature, our approach is based on a soft neck, since its motion capabilities are more convenient for this task, and provide a better compliance, safer human-robot interaction and adaptability to complex environments.

In this case, the head stabilization is achieved with a feedback control scheme based on the neck's embedded IMU device, providing full disturbance rejection and effective inclination control, very similar to behaviors observed in the nature. The system allows setting any head gaze and maintain it despite the robot's lower body positions or gait disturbances, in a way very similar to humans and other animals, which interestingly are also walkers.

Along with the advantages provided by the soft robotics approach, there are also many open challenges, like the problematic kinematic definition or the high non-linearity, which complicates modeling and makes these systems difficult to control. Following the decoupled approach described in [35] and [34], and a feedback control strategy, an efficient solution using PID and FOPID controllers is proposed.

In order to assess the system performance with both controllers, an extensive set of experiments was carried out, including step response and real humanoid motion tests, all of them under varying payload conditions. As expected, given the plant characteristics, the classic PID controllers provide a poor performance even using robust design constraints, resulting in oscillations and unstable behavior for different inclinations and payloads. However, with the robust fractional controller, the performance and stability are drastically improved. As the theory and experiments avail, the FOPI provides more flexibility and allows achieving more demanding control requirements, offering better results due to a more moderate phase contribution than the integer order operator-based controllers.

Reviewer 3 Report

Comments and Suggestions for Authors

In the present work, the authors have proposed a new approach for humanoid robot head camera stabilization, based on a bio-inspired soft neck. The neck is used to provide more compliant head movements, but presents a complex kinematic model and a non-linear behavior due to its soft nature, which requires a robust control solution. The results show the superiority of the fractional order control approach, which provides a higher performance robustness. The background is good as it deals with many industrial applications.

The work falls within the scope of the target journal. The logic of the work is well organized. The language is roughly smooth. The results sound credible. It can be considered for publication after some modifications.

1.      The reference format should be [7-9] not [7] [8] [9].

2.      The parameters of both sides of Eq. (1) are not consistent.

3.      How to derive the transfer functions?

4.      The authors may more concentrate on the forces induced by the robot.

5.      Are there any experimental verification for the theoretical model?

Comments on the Quality of English Language

none

Author Response

In the present work, the authors have proposed a new approach for humanoid robot head camera stabilization, based on a bio-inspired soft neck. The neck is used to provide more compliant head movements, but presents a complex kinematic model and a non-linear behavior due to its soft nature, which requires a robust control solution. The results show the superiority of the fractional order control approach, which provides a higher performance robustness. The background is good as it deals with many industrial applications.

Thank you very much for taking the time to carefully read and evaluate our manuscript. Your constructive comments and thoughtful suggestions have provided us with valuable insights that will significantly contribute to enhancing the quality and impact of our work.

The work falls within the scope of the target journal. The logic of the work is well organized. The language is roughly smooth. The results sound credible. It can be considered for publication after some modifications.

1. The reference format should be [7-9] not [7] [8] [9].

Thank you for pointing that out. Using the format [7-9] instead of listing each reference separately ([7][8][9]) is a clearer way to indicate consecutive citations within the same numbered range. We appreciate your attention to detail and changed the citation style as proposed.

2. The parameters of both sides of Eq. (1) are not consistent.

Thank you for bringing this to our attention. We agree that it's important for the parameters on both sides of an equation to be consistent in order for it to be valid. In order to solve the issue, we have included the required conversion parameters kθ and kÏ•, and introduced them in the paper with the following paragraph:

Given that link compression can be neglected, as neck geometry favors the lateral bending,

δ can be considered constant (in this case, δi = 0.12 m). The parameters kθ and kÏ• are

variable and depend on the current neck position. Since the proposed controllers are robust

to plant changes, they can be considered constant for modeling. Furthermore, given the

empirical approach, they can be integrated into the identified transfer function, and thus

can be defined as kθ = 1 rad/m and kÏ• = 1 rad/m without loss of generality.

3. How to derive the transfer functions?

As described in section 2.1 (System identification), the transfer functions of the system were derived using recursive least squares (RLS) identification algorithm. In this approach, the model is represented as a continuous-time transfer function, and the unknown coefficients are estimated recursively based on the input-output measurements collected during experiments.

In that direction, we have clarified the selection of the model order with the following paragraph:

After testing different modeling options, the most accurate result was obtained with the standard two-pole and gain control engineering model.

4. The authors may more concentrate on the forces induced by the robot.

Firstly, we would like to express our gratitude once again for your meticulous evaluation of our manuscript and your valuable feedback. After careful consideration of your suggestions, we regret to inform you that incorporating a stronger emphasis on the forces induced by the robot falls outside the current scope of our work. As you rightfully pointed out, our primary focus lies on proposing a novel approach for humanoid robot head camera stabilization, leveraging a bio-inspired soft neck mechanism. Expanding the scope to include force control would require substantial modifications, potentially detracting from the core objectives of the paper.

Additionally, we acknowledge your interest in exploring force control methodologies for our robotic system, and we remain open to future explorations of advanced force control techniques regarding system stability and robustness.

5. Are there any experimental verification for the theoretical model?

Yes, we performed a direct experimental verification during the experimental phase, however, these results were not included in the final version of the manuscript, since the modeling was already extensively discussed in the works [27] and [35]. In this case, we decided to mention those works, avoiding paper overload, and providing only essential information for the results' verification. Both studies show a high correspondence between theoretical predictions and experimental measurements.

On the other hand, the controller experiments provided in the paper, indirectly validate the model, since consistent patterns are observed between the predicted results and the actual measurements.

We have highlighted this information in the document through the addition of the following paragraph:

The resulting model shares many similarities with the one presented in [35], as both approaches share several fundamental assumptions about the underlying physics. In this case, as shown below in the experimental section, the simulations match the actual system behavior with high accuracy, validating the proposed model.

Once again, we deeply appreciate your keen eye for detail and your commitment to upholding high standards in scientific reporting. Please accept our assurance that the suggested changes were implemented promptly and comprehensively.